HU-EP-24/10

# On exactly solvable Yang-Baxter models and enhanced symmetries

Khalil Idiab and Stijn J. van Tongeren

*Institut für Mathematik und Institut für Physik, Humboldt-Universität zu Berlin,*
*IRIS Gebäude, Zum Grossen Windkanal 6, 12489 Berlin, Germany*

khalil.idiab@physik.hu-berlin.de // svantongeren@physik.hu-berlin.de

## Abstract

We study Yang-Baxter deformations of the flat space string that result in exactly solvable models, finding the Nappi-Witten model and its higher dimensional generalizations. We then consider the spectra of these models obtained by canonical quantization in light-cone gauge, and match them with an integrability-based Bethe ansatz approach. By considering a generalized light-cone gauge we can describe the model by a nontrivially Drinfel'd twisted S matrix, explicitly verifying the twisted structure expected for such deformations. Next, the reformulation of the Nappi-Witten model as a Yang-Baxter deformation shows that Yang-Baxter models can have more symmetries than suggested by the $r$ matrix defining the deformation. We discuss these enhanced symmetries in more detail for some trivial and nontrivial examples. Finally, we observe that there are nonunimodular but Weyl-invariant Yang-Baxter models of a type not previously considered.

# 1 Introduction

Integrable sigma models have played an important role in developing our detailed understanding of the AdS/CFT correspondence. With the development of Yang-Baxter deformations of superstring sigma models, the scope of integrability now extends to a large variety of deformed string theories with reduced symmetry, of direct relevance to the gauge/gravity correspondence.

Importantly, Yang-Baxter sigma models provide a setting to test existing as well as novel holographic dualities, while keeping access to the powerful tools of integrability. However, while most new types of deformations of e.g. the AdS$_5$ superstring are well understood at the classical level, their quantum integrable structure remains largely to be unveiled. In this paper we will study simple Yang-Baxter deformations of the flat space string, amenable to direct canonical quantization, which thereby provide a small but exact window into the quantum structure of Yang-Baxter sigma models.

The famous AdS$_5$ superstring appearing in the canonical example of AdS/CFT as the dual of maximally supersymmetric Yang-Mills theory, is described by a rather involved sigma model [1, 2]. Due to its nontrivial Ramond-Ramond background it cannot be directly approached via conventional CFT methods, and due to its complicated interaction terms, it also cannot be straightforwardly canonically quantized in light-cone gauge. Nevertheless, we now have a fantastic understanding of the spectrum of this string, building on its integrability [3]. Namely, under the assumption that integrability persists at the quantum level, the spectrum of the AdS$_5$ string can be described in terms of factorized scattering and the (thermodynamic) Bethe ansatz and quantum spectral curve, see e.g. the reviews [2,4,5]. Other observables such as Wilson loops and higher point functions can also be approached using integrability, see e.g. the recent [6, 7] and references therein.

Yang-Baxter deformations[1] of strings [10–14] give rise to a landscape of models with reduced symmetry and a variety of underlying algebraic structures. The deformation may even break the Weyl invariance of the string [15–17], but this is avoided if the $r$ matrix defining a given Yang-Baxter deformation is unimodular [18]. In terms of quantum integrability, inhomogeneous Yang-Baxter deformations ($q$ deformations) of the superstring can be tackled by the same light-cone gauge and exact S matrix methods as the undeformed string [19–23]. Homogeneous deformations on the other hand, come in a variety of types corresponding to different Drinfel'd twists [24–26]. In this setting, only abelian deformations based on Cartan generators fit directly with the undeformed approach, with the associated Drinfel'd twists naturally adapted to the undeformed light-cone gauge S matrix, as verified at tree level in [27]. While currently lacking an exact quantum description, other homogeneous deformations of the AdS string can however be studied at the semiclassical level through their classical spectral curve [28, 29], and may in the future prove accessible through the alternate light-cone gauge fixings recently studied in [30] at the undeformed level. In terms of AdS/CFT, homogeneous Yang-Baxter deformations of AdS strings are conjectured to be dual to twist-noncommutative deformations of the dual gauge theory [25, 26], and recently there has been significant progress on the explicit construction of such noncommutative deformations of 4D maximally supersymmetric Yang-Mills theory in particular [31, 32].

The first aim of this paper is to explicitly verify the Drinfel'd twisted structure of homogeneous deformations, by studying them in the simplified setting of the flat space string and looking for models that can be explicitly quantized in light-cone gauge. Concretely, we will look at the Yang-Baxter deformed flat space string constructed in [33], and find a class of $r$ matrices that results in a plane wave background, resulting in a quadratic model in light-cone gauge. Our

---

[1]Yang-Baxter sigma models as deformations of principal chiral models were originally introduced in [8] and their integrability shown in [9].

findings suggest that there is only one such class of Yang-Baxter models, equivalent to strings on the Nappi-Witten background and its higher dimensional generalizations. Focusing on the four dimensional case, we explicitly quantize the model in light-cone gauge, and match the resulting expression with a factorized scattering approach. By working in a generalized light-cone gauge, the expected effect of the deformation is an overall momentum shift combined with a particular Drinfel'd twist, and we show how these two effects combine to match the spectrum obtained through canonical quantization, verifying the Drinfel'd twisted structure of this model at the quantum level.

The second part of the paper starts from the observation that the Nappi-Witten model has more symmetries than naively expected from the Yang-Baxter perspective. The ten dimensional $\mathfrak{iso}(1,3)$ symmetry of $\mathbb{R}^{1,3}$ gets broken to a three dimensional abelian algebra, while from the Nappi-Witten perspective as a Wess-Zumino-Witten (WZW) model based on the centrally extended two dimensional Euclidean algebra, it is clear that the model should have a seven dimensional symmetry algebra. In other words, this particular Yang-Baxter deformation gives us an example where the background has enhanced symmetries, compared to those suggested by the $r$ matrix defining the deformation.[2] We discuss this mismatch in general terms, but do not have a conclusive criterion determining which Yang-Baxter deformations admit such enhanced symmetries. As further examples, we discuss higher dimensional Nappi-Witten type backgrounds, focussing on the six dimensional case in particular, and provide an overview of enhanced symmetries in all abelian rank two Yang-Baxter deformations of $\mathbb{R}^{1,3}$. Finally, we observe that, somewhat unexpectedly, also Weyl invariance can be enhanced, by discussing how flat space admits at least one non-unimodular deformation which is clearly Weyl invariant.

This paper is organized as follows. We start with a brief recap of the construction of Yang-Baxter sigma models in section 2. In section 3 we discuss the class of plane wave Yang-Baxter models whose spectra we study in section 4, illustrating their exact Drinfel'd twisted structure. Then in section 5 we discuss the alternate formulation of our plane wave models as Nappi-Witten type models, which leads us to the notion of enhanced symmetries which we discuss in section 6. We conclude with a number of open questions for further study, and give several appendices with technical details.

## 2 Yang-Baxter sigma models

Symmetric space sigma models and their Yang-Baxter deformations are an interesting class of two dimensional integrable models. In this section we will recall the general construction of the Yang-Baxter coset sigma model, and in the process fix our conventions. Our field theory lives on a two dimensional worldsheet denoted $\Sigma$ and the undeformed target space is a coset space $\mathcal{M} = G/H$. The groups $G$ and $H$ have Lie algebras $\mathfrak{g}$ and $\mathfrak{g}^{(0)}$ respectively, where $\mathfrak{g}$ is required

---

[2]A simpler example of this is the $r = p_1 \wedge p_2$ deformation of flat space, i.e. a TsT transformation in two Cartesian directions. This deformation does not actually change the geometry, and the resulting model is clearly maximally symmetric despite the apparent deformation. Our Nappi-Witten example as well as other cases we discuss, have a richer structure.

to have a grading corresponding to a symmetric space, namely

$$\mathfrak{g} = \mathfrak{g}^{(0)} \oplus \mathfrak{g}^{(1)}, \tag{2.1}$$

$$[\mathfrak{g}^{(0)}, \mathfrak{g}^{(0)}] \subset \mathfrak{g}^{(0)}, \qquad [\mathfrak{g}^{(0)}, \mathfrak{g}^{(1)}] \subset \mathfrak{g}^{(1)}, \qquad [\mathfrak{g}^{(1)}, \mathfrak{g}^{(1)}] \subset \mathfrak{g}^{(0)}. \tag{2.2}$$

To construct an action we need a nondegenerate symmetric bilinear form on $\mathfrak{g}$, $\langle \cdot | \cdot \rangle$, which needs to be grade compatible

$$\langle X | \mathcal{P} Y \rangle = \langle \mathcal{P} X | \mathcal{P} Y \rangle, \tag{2.3}$$

where $\mathcal{P} : \mathfrak{g} \to \mathfrak{g}^{(1)}$ is the projector onto the grade one subspace of $\mathfrak{g}$, and $\mathrm{Ad}_H$ invariant

$$\langle X | Y \rangle = \langle h^{-1} X h | h^{-1} Y h \rangle, \qquad \forall h \in H. \tag{2.4}$$

Our worldsheet is parameterized by coordinates $\sigma^0 = \tau, \sigma^1 = \sigma$, its cotangent space $T^* \Sigma$ is spanned by $\{ d\sigma^\alpha \}$, and we denote the worldsheet metric by $h_{\alpha\beta}$. The Yang-Baxter deformed symmetric space sigma model action is now written in terms of the Maurer-Cartan one form $A = -g^{-1} dg$ as

$$S[g] = \frac{1}{2} \int_\Sigma \langle A | \star \mathcal{P} D A \rangle, \tag{2.5}$$

where the wedge product is implicitly included in the inner product, and the deformation operator $D$ is defined as

$$D = \frac{1}{1 + \eta R_g \mathcal{P} \star}, \qquad\qquad R_g(X) = r^{ij} \langle g^{-1} T_i g | X \rangle g^{-1} T_j g, \tag{2.6}$$

where $\{ T_i \}$ forms some basis for $\mathfrak{g}$ and $\star$ denotes the Hodge dual. We will frequently refer to the $R$ operator above in the form of its associated $r$ matrix $r = r^{ij} T_i \wedge T_j \in \Lambda^2(\mathfrak{g})$, where $a \wedge b = (a \otimes b - b \otimes a)/2$.

The equations of motion of this model can be written in terms of the deformed current $I = DA$ as

$$d \star \mathcal{P} I - [I, \star \mathcal{P} I] = 0, \tag{2.7}$$

where the commutator of forms includes an implicit wedge product as well, i.e. $[A, B] = A \wedge B + B \wedge A = \epsilon^{\alpha\beta} [A_\alpha, B_\beta] d\tau \wedge d\sigma$. The deformed current $I$ is flat on-shell provided that the $R$ operator is antisymmetric, i.e. $r^{ij} = -r^{ji}$, and solves the CYBE

$$\langle [R_g X, R_g Y] | Z \rangle + \langle [R_g Z, R_g X] | Y \rangle + \langle [R_g Y, R_g Z] | X \rangle = 0, \qquad X, Y, Z \in \mathfrak{g}. \tag{2.8}$$

In this case we can find a Lax connection with the following ansatz

$$L(z) = I + \ell_1(z) \mathcal{P} I + \ell_2(z) \star \mathcal{P} I, \tag{2.9}$$

where in the semi-simple setting $\ell_1$ and $\ell_2$ have to satisfy $\ell_2^2 - \ell_1^2 - 2\ell_1 = 0$ with $\ell_2 \neq 0$. Beyond the semi-simple setting, solutions to the *inhomogeneous* CYBE need to be treated on a case by case basis. Moreover, homogeneous deformations of flat space have more freedom, allowing us to set $\ell_1 = 0$, keeping $\ell_2$ itself as the spectral parameter. For further details we refer to e.g. [34, 33].

## 2.1 Coordinate representation

Introducing a coordinate system on the coset space in the form of a coset representative allows us to write the action in Polyakov form

$$S[x] = \frac{1}{2} \int_\Sigma d^2\sigma \left( \sqrt{h} h^{\alpha\beta} - \epsilon^{\alpha\beta} \right) (G_{\mu\nu} + B_{\mu\nu}) \, \partial_\alpha x^\mu \partial_\beta x^\nu, \tag{2.10}$$

where $h = |\det h_{\alpha\beta}|$ and

$$G + B = \left( g^{-1} + \eta r \right)^{-1}, \tag{2.11}$$

$r$ denotes the matrix in the Killing vector representation – $r^{\mu\nu} = r^{ij} \chi_i^\mu \chi_j^\nu$, with $\chi_i$ the Killing vector associated to the generator $T_i$ – and $g$ denotes the undeformed metric $g_{\mu\nu} = \langle T_i | \mathcal{P} T_j \rangle A_\mu^i A_\nu^j$ defined through the components of the Maurer-Cartan form $A = A_\mu^i T_i dx^\mu$. For light-cone gauge fixing it is convenient to present the action in first order formalism,

$$S[x, p] = \frac{1}{2} \int_\Sigma d^2\sigma \left( p_\mu \dot{x}^\mu + \frac{h^{01}}{h^{00}} p_\mu x'^\mu - \frac{1}{2\sqrt{h} h^{00}} C \right), \tag{2.12}$$

$$C = G^{\mu\nu} \left( p_\mu + B_{\mu\rho} x'^\rho \right) \left( p_\nu + B_{\nu\lambda} x'^\lambda \right) + G_{\mu\nu} x'^\mu x'^\nu, \tag{2.13}$$

see for example [2]. Using (2.11), for Yang-Baxter deformations the expression for $C$ takes a simple form in terms of the undeformed metric and $r$ matrix,

$$C = g^{\mu\nu} p_\mu p_\nu + g_{\mu\nu} \left( x'^\mu + \eta r^{\mu\rho} p_\rho \right) \left( x'^\nu + \eta r^{\nu\lambda} p_\lambda \right). \tag{2.14}$$

# 3 Plane wave Yang-Baxter deformations

An important category of gravitational backgrounds are so called gravitational pp-waves. They are given by a metric of the form

$$ds^2 = K(x^+, \vec{x})(dx^+)^2 - 2dx^+(dx^- + A_i(x^+, \vec{x})dx^i) + g_{ij} dx^i dx^j \tag{3.1}$$

together with a possibly nontrivial $B$ field $B_{\mu\nu} dx^\mu \wedge dx^\nu$. For the choice of $g_{ij} = g_{ij}(x^+)$, $A_i = \frac{1}{2} A_{ij}(x^+) x^j$, $B_{i+} = \frac{1}{2} b_{ij}(x^+) x^j$, they provide a well-known class of string backgrounds, with one loop Weyl invariance requiring

$$\partial_i \partial^i K = \frac{1}{2} \left( A_{ij} A^{ij} - b_{ij} b^{ij} \right). \tag{3.2}$$

It is also well known that if $K$ is quadratic in the $x_i$ these backgrounds lead to exactly solvable sigma models, as they become quadratic in the transverse fields upon fixing a light-cone gauge. In AdS/CFT in particular, an important role is played by gravitational waves with $K = \sum_{i=1}^n (x_i)^2$, $A = 0$, $g_{ij} = \delta_{ij}$.[3] Motivated by their exact solvability and relevance in AdS/CFT, we would like to understand whether such exactly solvable plane wave backgrounds can arise as Yang-Baxter deformations of the flat space string. In Appendix A we discuss the constraints on the $r$ matrix to obtain a quadratic Hamiltonian, and more specifically particular plane wave backgrounds, starting from flat space.

---

[3]In the case of AdS$_3$ it is possible to support this background by a nontrivial NSNS flux, cf. eqn. (3.2), while in other cases the role of the NSNS form is taken over by the RR forms.

## 3.1 Plane wave Yang-Baxter deformations

Focusing on the simplest case with $K$ quadratic, $A = 0$, and no explicit $x^+$ dependence (no time dependence in light-cone gauge), this means that we are looking for a metric of the form

$$ds^2 = \omega_{ij} x^i x^j (dx^+)^2 - 2dx^+ dx^- + dx_i dx^i, \tag{3.3}$$

where $g_{ij}$ has been brought to canonical $\delta_{ij}$ form as is always possible in this case. Compared to the Yang-Baxter background, if we want to get this type of background on the nose,[4] as discussed in Appendix A.1 we are looking for Yang-Baxter deformations with only $r^{-i}(\vec{x})$ nonzero, and at most linear in the transverse fields $r^{-i} = c^i_j x^j$. Explicitly finding all $r$ matrices satisfying this constraint, solving the classical Yang-Baxter equation, and giving a Weyl-invariant sigma model, is a nontrivial question. In four dimensions, we fortunately have a classification of $r$ matrices available [35], and the only $r$ matrix with at most nonzero $r^{-i}$ and linear coordinate dependence is

$$r = p_- \wedge m_{23}. \tag{3.4}$$

Here and below the $p_\mu$ denote the translation generators of the Poincaré algebra, with Killing vector representation $\partial_\mu$, and $m_{\mu\nu}$ the Lorentz generators with Killing vector representation $x_\mu \partial_\nu - x_\nu \partial_\mu$. We included non-unimodular $r$ matrices in this analysis, because unimodularity is not strictly required for Weyl invariance as we will come back to in section 6.6.

In higher dimensions the general problem quickly becomes practically untractable, but by brute force evaluation of the background constraints and the CYBE we were able to show that for $r$ matrices of up to rank 6 (three independent wedge terms) in $\mathfrak{iso}(1,9)$ there are no new solutions, except the obvious multi-parameter generalization of the $r$ matrix (3.4)

$$r = p_- \wedge (\alpha m_{23} + \beta m_{45} + \ldots), \tag{3.5}$$

i.e. $r = p_- \wedge c$ with $c$ an arbitrary element of the Cartan subalgebra of the transverse rotational $SO(d-1)$ symmetry in arbitrary dimension $d$.

The background associated to $r$ matrix (3.4) is

$$ds^2 = -\eta^2 (x_2^2 + x_3^2)(dx^+)^2 - 2dx^+ dx^- + dx_i dx^i,$$
$$B = \eta \left( x^3 dx^2 - x^2 dx^3 \right) \wedge dx^+, \tag{3.6}$$

while its higher dimensional counterpart associated to the $r$ matrix (3.5) is

$$ds^2 = -\sum_{i=1}^{n} \eta_i^2 (x_{2i}^2 + x_{2i+1}^2)(dx^+)^2 - 2dx^+ dx^- + dx_i dx^i,$$
$$B = \sum_{i=1}^{n} \eta_i \left( x^{2i+1} dx^{2i} - x^{2i} dx^{2i+1} \right) \wedge dx^+. \tag{3.7}$$

The background (3.6) is a particularly well-known plane wave background, corresponding to the Nappi-Witten model [36], as we will come back to in detail later.

---

[4]Considering this question up to diffeomorphisms instead, is impractical to answer at the purely geometric level. Answering it algebraically through constraints such as preserving a null Killing vector in the Yang-Baxter model context (via symmetries of the $r$ matrix) seems promising a priori, but we will later see that Yang-Baxter backgrounds may have more symmetries than suggested by the $r$ matrix, meaning such an approach would not automatically be exhaustive either.

# 4 Exact spectra and Drinfel'd twists

We would now like to discuss the exact solvability of strings on the background (3.6), and explain the resulting spectrum in terms of integrability, in particular in terms of Drinfel'd twists expected to arise in homogeneous Yang-Baxter models. We will focus on the four dimensional part of the model that is actually deformed, dropping the standard contributions from undeformed transverse directions. The spectra for the higher dimensional models of eqs. (3.7) follow analogously.

## 4.1 Canonical quantization

In this section we aim to find the energy spectrum of the flat space deformation associated to $r = p_- \wedge m_{23}$ with background (3.6), or equivalently, the Nappi-Witten model. This spectrum has been previously determined in [37], here we independently derive it in convenient conventions for comparison to an integrability-based approach. We will use a coordinate system $x^\mu = (x^+, x^-, x, \overline{x})$ where $x = \frac{1}{\sqrt{2}} \left( x^2 + ix^3 \right)$ and $\overline{x} = \frac{1}{\sqrt{2}} \left( x^2 - ix^3 \right)$, related by complex conjugation for reality. The light-cone gauge worldsheet Hamiltonian is given by (A.2) and comes out to be

$$\mathcal{H}_{\mathrm{ws}} = p\bar{p} + x'\overline{x}' + \eta^2 x\overline{x} - i\eta \left( \overline{x}x' - x\overline{x}' \right), \tag{4.1}$$

with $\eta$ as the deformation parameter. The equations of motion separate into

$$\ddot{x} - x'' + \eta^2 x - 2i\eta x' = 0, \tag{4.2}$$

and the complex conjugate equation for $\bar{x}$. Since the classical EOM are linear we can find periodic solutions $x(\tau, \sigma) = x(\tau, \sigma + 2\pi R)$ with the ansatz

$$x(\tau, \sigma) = \sum_{n=-\infty}^{\infty} \left( a_n^+ e^{i\omega_n \tau} + a_n^- e^{-i\omega_n \tau} \right) e^{in\sigma/R}, \tag{4.3}$$

$$\overline{x}(\tau, \sigma) = \sum_{n=-\infty}^{\infty} \left( \overline{a}_n^+ e^{-i\omega_n \tau} + \overline{a}_n^- e^{i\omega_n \tau} \right) e^{-in\sigma/R}, \tag{4.4}$$

with $\overline{a}_n^\pm = (a_n^\pm)^*$ and $\omega_n = n/R + \eta$. The Virasoro constraint $p_\mu x'^\mu = 0$ yields the level matching constraint by imposing that $x^-$ should also be periodic, taking the form

$$\int_0^{2\pi R} \left( \dot{\overline{x}}x' + \dot{x}\overline{x}' \right) d\sigma = 0. \tag{4.5}$$

Plugging in the mode expansion we find

$$\sum_{n=-\infty}^{\infty} \omega_n n \left( \overline{a}_n^+ a_n^+ - \overline{a}_n^- a_n^- \right) = 0. \tag{4.6}$$

Similarly we can find the worldsheet energy in terms of oscillators

$$H_{\mathrm{ws}} = \int_0^{2\pi R} \mathcal{H}_{\mathrm{ws}} d\sigma = 4\pi R \sum_{n=-\infty}^{\infty} \omega_n^2 \left( \overline{a}_n^+ a_n^+ + \overline{a}_n^- a_n^- \right). \tag{4.7}$$

In order to canonically quantize we need the Poisson brackets of the oscillators, which are induced by the canonical brackets

$$\{p(\tau, \sigma), x(\tau, \sigma')\} = \delta(\sigma - \sigma'), \tag{4.8}$$

$$\{\bar{p}(\tau, \sigma), \bar{x}(\tau, \sigma')\} = \delta(\sigma - \sigma'). \tag{4.9}$$

The nonvanishing brackets are

$$\{a_n^\pm, \bar{a}_{n'}^\pm\} = \pm \frac{\delta_{nn'}}{4\pi i \omega_n R}. \tag{4.10}$$

We can now canonically quantize with these brackets, worldsheet energy and level matching condition as carried out in Appendix B. Up to a normal ordering constant, the energy spectrum is given by

$$E_{\{N, \bar{N}\}} = \sum_{n=-\infty}^{\infty} \left| \frac{n}{R} + \eta \right| \left( N_n + \tilde{N}_n \right) \tag{4.11}$$

where $\{N_n, \tilde{N}_n\}$ are all non-negative integers subject to the level matching condition

$$\sum_{n=-\infty}^{\infty} n \left( N_n - \tilde{N}_n \right) = 0. \tag{4.12}$$

This spectrum matches the result from [37] up to the normal ordering constant which can be found there. Interestingly, for sufficiently small deformations, $-1 \leq \eta R \leq 1$, as shown in Appendix C the possible energy levels are given by

$$E(k, \ell) = \frac{2k}{R} + \ell|\eta|, \tag{4.13}$$

labeled by integers $k \geq 0$ and $\ell \geq -2k$.

## 4.2 Spectrum from Bethe ansatz

We want to use the above results on the spectrum to perform a check of the typical integrability-based approach to Yang-Baxter models, relying on exact S matrices and the Bethe ansatz. We expect a general homogeneous deformation to enter an undeformed model through a Drinfel'd twist associated to the $r$ matrix defining the deformation. However, the deformation we are considering is special, and at first glance appears too simple to see this structure. Namely, a $p_- \wedge m_{23}$ deformation takes the undeformed model, and simply shifts the momentum in its description uniformly by a term proportional to the $m_{23}$ charge of the relevant particle [27]: $p \to p \pm \eta$, as e.g. in $\omega_n$ of the previous section. Since in the standard light-cone gauge, the undeformed worldsheet theory is free, we start from a trivial S matrix, and our deformed model is simply described by trivial Bethe equations for the two types of excitations associated to the $x$ and $\bar{x}$ fields of our gauge fixed model of section 4.1. I.e. we have

$$e^{2\pi i R p_k} = 1, \qquad e^{2\pi i R \bar{p}_n} = 1, \qquad \forall k, n. \tag{4.14}$$

while the effect of the deformation is entirely contained in the shifted dispersion relations $\omega(p) = |p - \eta|$ and $\omega(\bar{p}) = |\bar{p} + \eta|$. These equations are now solved by the usual

$$2\pi R p_k = 2\pi n_k, \tag{4.15}$$

$$2\pi R \bar{p}_k = 2\pi \bar{n}_k, \tag{4.16}$$

with $n_k, \bar{n}_k \in \mathbb{Z}$, giving a worldsheet energy

$$H_{\text{w.s.}} = \sum_{k=1}^{M} \omega(p_k) + \sum_{\bar{k}=1}^{\overline{M}} \omega(\bar{p}_{\bar{k}}) = \sum_{k=1}^{M} \left| \frac{n_k}{R} - \eta \right| + \sum_{\bar{k}=1}^{\overline{M}} \left| \frac{\bar{n}_{\bar{k}}}{R} + \eta \right|. \tag{4.17}$$

The level matching condition remains undeformed

$$L = \sum_{k=1}^{M} p_k + \sum_{\bar{k}=1}^{\overline{M}} \bar{p}_{\bar{k}} = \sum_{k=1}^{M} \frac{n_k}{R} + \sum_{\bar{k}=1}^{\overline{M}} \frac{\bar{n}_{\bar{k}}}{R} = 0. \tag{4.18}$$

To match the spectrum obtained from canonical quantization (4.11), we rewrite the sums using

$$\sum_{k=1}^{M} f(n_k) = \sum_{n=-\infty}^{\infty} f(n) A_n, \qquad \sum_{\bar{k}=1}^{\overline{M}} f(\bar{n}_{\bar{k}}) = \sum_{n=-\infty}^{\infty} f(n) B_n, \tag{4.19}$$

which can always be done for $n_k, \bar{n}_{\bar{k}} \in \mathbb{Z}$ and $A_n, B_n \in \mathbb{N}^0$. With these, the energy and level matching condition become

$$H_{\text{w.s.}} = \sum_{n=-\infty}^{\infty} \left| \frac{n}{R} - \eta \right| A_n + \sum_{n=-\infty}^{\infty} \left| \frac{n}{R} + \eta \right| B_n, \tag{4.20}$$

$$L = \sum_{n=-\infty}^{\infty} \frac{n}{R} (A_n + B_n) = 0. \tag{4.21}$$

Now we simply identify $A_n = N_{-n}$, $B_n = \tilde{N}_n$, to match the result from canonical quantization of the previous subsection.[5] Next, we would like to change perspectives slightly, in order to manifest the Drinfel'd twisted structure that does appear in this model.

## 4.3 Drinfel'd twisted S matrix

We can manifest more of the structure of our deformation by changing our gauge. Let us introduce generalized light cone coordinates [2] of the form

$$\hat{x}^+ = x^+ + \frac{1}{2} \alpha x^-, \qquad \hat{x}^- = x^-, \tag{4.22}$$

with conjugate momenta

$$\hat{p}_+ = p_+, \qquad \hat{p}_- = p_- - \alpha p_+. \tag{4.23}$$

Instead of our previous gauge choice $x^+ = \tau, p_- = 1$, we now fix

$$\hat{x}^+ = \tau, \qquad \hat{p}_- = 1, \tag{4.24}$$

---

[5]In our canonical quantization discussion we did not determine the normal ordering constant. In the present integrability-based approach, this constant would follow by including wrappping corrections through the (mirror) thermodynamic Bethe ansatz (TBA), instead of the asymptotic Bethe ansatz that we used. While the theory is free so that there are no interaction kernels in the TBA, there are still nontrivial but simple wrapping corrections, which for our simple type of theory lead to a constant shift in the spectrum, see [38] section 4 and Appendix E for a closely related discussion. We thank A. Sfondrini for discussions on this point.

which gives

$$H_{\mathrm{ws}} = -P_+, \qquad \int_0^{2\pi R(\alpha)} \hat{p}_- d\sigma = 2\pi R(\alpha) = P_- - \alpha P_+, \qquad (4.25)$$

with $P_\pm = \int_0^{2\pi R} p_\pm d\sigma$. $\alpha$ labels a space of different gauge choices, which should each give the same physical spectrum. For $\alpha = 0$ we are back at the standard light-cone gauge, where the undeformed worldsheet Hamiltonian is quadratic and the S matrix is trivial. The effect of nonzero $\alpha$ on the S matrix is well known [39, 2], see also [40], and in this case means[6]

$$S(p_k, p_j; \alpha) = e^{i\alpha(p_j\omega_k - p_k\omega_j)}, \qquad (4.26)$$

where we have collected the momenta of both types of excitations in a set $\{p_k\}$ with a single label $k$. This $\alpha$ dependence is consistent with the Bethe ansatz equations

$$e^{2\pi i R(\alpha)p_k} \prod_{j \neq k} S(p_k, p_j; \alpha) = 1, \qquad \forall k, \qquad (4.27)$$

reducing to the $\alpha$-independent[7]

$$e^{i2\pi R p_k} = 1, \qquad \forall k, \qquad (4.28)$$

where $R = R(0) = P_-/(2\pi)$.

From this new perspective, the $r$ matrix defining our deformation looks like

$$r = p_- \wedge m_{23} = (\hat{p}_- + \alpha\hat{p}_+) \wedge m_{23}. \qquad (4.29)$$

We still get a momentum shift from the $\hat{p}_- \wedge m_{23}$ term, but now also get a second contribution from the $\hat{p}_+ \wedge m_{23}$ term. The latter is expected to deform the $S$ matrix by a Drinfel'd twist of the form

$$S \to e^{-i\alpha\hat{p}_+ \wedge m_{23}} S e^{-i\alpha\hat{p}_+ \wedge m_{23}} \qquad (4.30)$$

Note that the Drinfel'd twist is $\alpha$ dependent, while the momentum shift is not. Denoting the $m_{23}$ charge of the $k$th Bethe ansatz particle by $m_k$, in total we then expect the worldsheet S matrix of our model to take the form

$$S_{\mathrm{def}}(p_k, p_j; \alpha) = e^{-i\alpha\hat{p}_+ \wedge m_{23}} S(p_k + m_k, p_j + m_j; \alpha) e^{-i\alpha\hat{p}_+ \wedge m_{23}}, \qquad (4.31)$$

where in our gauge, $-\hat{p}_+$ reads off the worldsheet energy $\omega$ of a given particle.

We would like to verify that this S matrix matches with our previous discussion, i.e. that precisely the expected Drinfel'd twist is required. Since the momentum shift is independent of $\alpha$, the dispersion relation is independent of $\alpha$, and to get Bethe equations that are independent of $\alpha$, we need the S matrix to take the undeformed $\alpha$-dependent form (4.26), except now with

---

[6]This type of S matrix is famously associated to the $T\bar{T}$ deformation [41, 42], which is no surprise given the relation between this deformation and generalized light-cone gauge fixing [43, 40]. For the flat space string this type of S matrix was originally discussed in [44]. Independent from the deformations considered in this paper, from a suitable perspective the $T\bar{T}$ S matrix itself can also be viewed as (arising from) a Drinfel'd twist [45].

[7]To explicitly see this, write out the Bethe equations as $e^{i(P_- + \alpha H_{\mathrm{ws}})p_k} \prod_{j \neq k} e^{i\alpha(p_j\omega_k - p_k\omega_j)} = 1$. Now use the level matching condition $\sum_j p_j = 0$ and the expression for the total energy $\sum_j \omega_j = H_{\mathrm{ws}}$, to rewrite $\prod_{j \neq k} e^{i\alpha(p_j\omega_k - p_k\omega_j)} = e^{i\alpha(\sum_{j \neq k} p_j\omega_k - p_k \sum_{j \neq k} \omega_j)} = e^{i\alpha(-p_k\omega_k - p_k(H_{\mathrm{ws}} - \omega_k))} = e^{-i\alpha p_k H_{ws}}$.

a shifted dispersion relation of course. Fortunately, this is indeed exactly the case, since the momentum shift outside the dispersion relation, and the twist conspire to exactly cancel in the S matrix,

$$S_{\text{def}}(p_k, p_j; \alpha) = e^{-\frac{1}{2}i\alpha(\omega_k m_j - \omega_j m_k)} S(p_k + m_k, p_j + m_j; \alpha) e^{-\frac{1}{2}i\alpha(\omega_k m_j - \omega_j m_k)}, \tag{4.32}$$

$$= e^{\frac{1}{2}i\alpha(\omega_j m_k - \omega_k m_j)} e^{i\alpha(p_j \omega_k + m_j \omega_k - p_k \omega_j - m_k \omega_j)} e^{\frac{1}{2}i\alpha(\omega_j m_k - \omega_k m_j)}, \tag{4.33}$$

$$= e^{i\alpha(p_j \omega_k - p_k \omega_j)}. \tag{4.34}$$

In summary, the effect of the deformation is the momentum shift appearing directly in the dispersion relation only, with the explicit momentum shift in the S matrix effectively cancelled precisely by the expected and required Drinfel'd twist.

# 5 Nappi-Witten model

The background (3.6) is a particularly well-known plane wave background, corresponding to the Nappi-Witten model [36]. From this perspective the background is associated to a WZW model based on the non-semi-simple centrally extended two dimensional Euclidean group. This group is generated by $P_1$, $P_2$, $J$ and the central element $T$, with nonzero Lie brackets

$$[J, P_i] = \epsilon_i{}^j P_j, \qquad [P_i, P_j] = \epsilon_{ij} T. \tag{5.1}$$

The Killing form on this algebra is degenerate, but it admits an alternate symmetric invariant bilinear form [36], with nonzero

$$\langle P_i | P_j \rangle = \delta_{ij}, \quad \langle J | J \rangle = b, \quad \langle J | T \rangle = 1, \tag{5.2}$$

where $b$ is an arbitrary constant. In our setting it is convenient to parameterize the group element as[8]

$$g = e^{\eta x^+ J} e^{x^i P_i} e^{\eta x^+ J} e^{-\left(\frac{x^-}{2\eta} + \eta b x^+\right) T}. \tag{5.3}$$

Evaluating the current $A = g^{-1} dg$ and substituting in the WZW action

$$S = \frac{1}{4\pi} \int_\Sigma d^2\sigma \langle A_\alpha | A^\alpha \rangle + \frac{i}{12\pi} \int_B d^3\sigma \epsilon_{\alpha\beta\gamma} \left\langle [A^\alpha, A^\beta] \middle| A^\gamma \right\rangle, \tag{5.4}$$

where B is a three manifold with boundary the worldsheet $\Sigma$, gives a sigma model on the background

$$ds^2 = -\eta^2(x_2^2 + x_3^2)(dx^+)^2 - 2dx^+ dx^- + dx_i dx^i,$$
$$B = \eta \left( x^3 dx^2 - x^2 dx^3 \right) \wedge dx^+, \tag{5.5}$$

i.e. exactly the 4D Yang-Baxter model background (3.6), where we used the total derivative freedom in $B$ for a precise match. This model is conformal to all orders in $\alpha'$, with central charge $c = 4$ [36]. From the perspective of the WZW action (5.4) we expect the symmetry algebra to be 7 dimensional, spanned by left and right versions of $P_1, P_2$ and $J$, and a shared central element $T$.[9] Before discussing these symmetries in more detail, let us discuss a relevant generalization of this model.

---

[8]While the Nappi-Witten model can be readily worked out using the abstract algebra structure alone, a matrix representation of the Nappi-Witten algebra can be useful in computer algebra applications, and we have included one in Appendix D.

[9] The Nappi-Witten model has been previously studied as a starting point for Yang-Baxter deformations in [46], where the authors found that Yang-Baxter deformations could only change the coefficient of the $B$ field.

## 5.1 Extended Nappi-Witten models

The background (3.7) corresponding to the higher dimensional $r$ matrix (3.5) can also be associated to a WZW model of Nappi-Witten type. We simply take the centrally extended two dimensional Euclidean algebra, and copy its momentum sector $n$ times, labeling them $P_i^{(k)}$, $k = 1, \ldots, n$, with commutation relations[10]

$$[J, P_i^{(a)}] = \eta_a \epsilon_{ij} P_j^{(a)}, \qquad [P_i^{(a)}, P_j^{(b)}] = \eta_a \delta^{ab} \epsilon_{ij} T, \tag{5.6}$$

where we have introduced $n$ distinct constants (deformation parameters) $\eta_a$ in the algebra. Similarly to the original Nappi-Witten case, this algebra admits an invariant symmetric bilinear form, given by

$$\left\langle P_i^{(a)} \middle| P_j^{(b)} \right\rangle = \delta^{ab} \delta_{ij}, \quad \langle J|J \rangle = b, \quad \langle J|T \rangle = 1, \tag{5.7}$$

where again $b$ is a constant. Considering a WZW model on the corresponding simply connected group gives a conformally invariant sigma model to all orders in $\alpha'$ – as for the original Nappi-Witten model – in $d = 2n + 2$ with central charge $c = 2n + 2$. With a group element of the form

$$g = e^{x^+ J} e^{\sum_{k=1}^{n} x^{i+2k} P_i^{(k)}} e^{x^+ J} e^{-\left(\frac{x^-}{2} + bx^+\right) T}, \tag{5.8}$$

the corresponding background is exactly the one of eqn. (3.7), again up to a total derivative in the $B$ field. For the particular case of $n = 2$ for example, we find the 6D plane wave background

$$ds^2 = -\eta_1^2 (x_2^2 + x_3^2)(dx^+)^2 - \eta_2^2(x_4^2 + x_5^2)(dx^+)^2 - 2dx^+ dx^- + dx_i dx^i, \tag{5.9}$$

$$B = \eta_1 \left( x^3 dx^2 - x^2 dx^3 \right) \wedge dx^+ + \eta_2 \left( x^5 dx^4 - x^4 dx^5 \right) \wedge dx^+. \tag{5.10}$$

For equal deformation parameters, this model appeared previously in section 2.2 of [49].

# 6  Enhanced symmetry in Yang-Baxter models

The Nappi-Witten model is known to be $O(d, d)$ dual to flat space [50,51], matching our current picture of it as an abelian Yang-Baxter deformation, i.e. a TsT transformation [52], of flat space. More interesting from the Yang-Baxter perspective, however, is the fact that the Nappi-Witten background has a seven dimensional isometry algebra[11], while only three of the original ten isometries of flat space survive the Yang-Baxter deformation – the $r$ matrix only commutes with $p_\pm$ and $m_{23}$. It appears that Yang-Baxter models, at least for flat space with its non-semi-simple isometry algebra, can have enhanced symmetry. In this section we will explore this in some detail.

---

This might appear to be at odds with our results showing that there is a Yang-Baxter deformation taking the Nappi-Witten model to flat space. However, [46] considered only left Yang-Baxter deformations, while in section 6.3 we will see that the deformation we are considering mixes the left and right symmetries of the Nappi-Witten model. It is also relevant to note that in apparent contrast to the results of [46], another group found that there is a nontrivial inhomogeneous Yang-Baxter deformation interpolating between the Nappi-Witten model and flat space [47]. We will come back to this point below.

[10]This is an example of a Nappi-Witten algebra, see [48], eqs. (B.6) and (B.7), matching directly if we choose $b = 0$ in our bilinear form.

[11]Left and right transformations, with the central elements of the two identified.

## 6.1 Noether symmetries and Killing vectors

Noether symmetries corresponds to off-shell infinitesimal symmetries of the action. For the Lagrangian of the deformed model (2.5), $\mathcal{L} = \frac{1}{2} \langle A | \star \mathcal{P}I \rangle$, we consider the general field transformation

$$g \to g' = kg = (1 + \epsilon) g, \qquad \epsilon \in \mathfrak{g}. \tag{6.1}$$

While the symmetries of the undeformed model correspond to constant $\epsilon$, in general this is not a requirement, and will turn out not to be the case in our setting. The variation of the Lagrangian is now

$$\delta \mathcal{L} = - \langle g^{-1} d\epsilon g | \star \mathcal{P}I \rangle - \eta \langle \mathcal{P}I | [R_g \mathcal{P}I, g^{-1}\epsilon g] \rangle. \tag{6.2}$$

To find conserved charges we need to find local $\epsilon$ such that the variation of the Lagrangian is at most a total derivative

$$\langle g^{-1} d\epsilon g | \star \mathcal{P}I \rangle + \eta \langle \mathcal{P}I | [R_g \mathcal{P}I, g^{-1}\epsilon g] \rangle = dC. \tag{6.3}$$

Analyzing the solution space of this equation is complicated in general, but there is a simple class of well-known solutions corresponding to manifest symmetries of the $R$ operator. Namely, if we consider the action written in terms of the undeformed currents and the $R_g$ operator, $\mathcal{L} = \frac{1}{2} \langle A | \star \mathcal{P} \frac{1}{1 + \eta R_g \mathcal{P}_\star} A \rangle$, it is clear that constant left multiplication of $g$, $g \to kg$, is a symmetry of the action with $C = 0$, provided $k \in G$ is a symmetry of the $R$ operator, meaning

$$R_{kg} = R_g. \tag{6.4}$$

In terms of the $r$ matrix, see eqn. (2.6) and following text, the sugroup $K \subset G$ of these transformations is generated by the generators $t$ which are symmetries of the $r$ matrix, i.e.

$$\Delta(\mathrm{ad}_t)(r) = (\mathrm{ad}_t \otimes 1 + 1 \otimes \mathrm{ad}_t)(r) = 0. \tag{6.5}$$

We will refer to these symmetries as manifest symmetries of the Yang-Baxter model.[12] Other solutions of eqn. (6.3) – those not arising via manifest symmetries of the $r$ matrix – we will refer to as enhanced symmetries.

It is illuminating to discuss these symmetries from a geometric perspective as well. For this, we start from the Yang-Baxter model background

$$G + B = \frac{1}{g^{-1} + \eta r}. \tag{6.6}$$

The full symmetry algebra of the undeformed model is geometrically realized by Killing vectors $\xi$ of the undeformed metric

$$\mathcal{L}_\xi(g) = 0, \tag{6.7}$$

---

[12]Alternatively, these solve eqn. (6.3) as follows. Consider an infinitesimal version corresponding to a constant $\epsilon$ that solves eqn. (6.5). Such $\epsilon$ now solve $\langle \mathcal{P}I | [R_g \mathcal{P}I, g^{-1}\epsilon g] \rangle = 0$, i.e. (6.3) for constant $\epsilon$ and $C = 0$, since we can use eqs. (6.4) and (6.5) (recall also eqn. (2.6)) to cancel the two combinations appearing in the implicit wedge product in the bilinear form, against each other.

where $\mathcal{L}_\xi$ denotes the Lie derivative along $\xi$. Now symmetries of the $r$ matrix as in (6.5) give rise to Killing vectors $\xi_t$ that do not just leave the metric invariant, but also the $r$ matrix,

$$\mathcal{L}_{\xi_t}(r) = 0. \tag{6.8}$$

By the product rule, such $\xi_t$ leave the full Yang-Baxter model background invariant. If we are looking for symmetries of a closed string sigma model, however, we only need the $B$ field to remain invariant up to a total derivative. In other words, the full set of symmetries of a Yang-Baxter model is generated by those $\xi$ for which

$$\mathcal{L}_\xi (G + B) = dC. \tag{6.9}$$

Note that any such $\xi$ with $\mathcal{L}_\xi B = 0$, by eqn. (6.6) leaves the undeformed metric and the $r$ matrix invariant, and hence corresponds to a manifest symmetry. In this language, the enhanced symmetries are nontrivial $\xi$ with $\mathcal{L}_\xi B \neq 0$. They do not correspond to symmetries of the $r$ matrix, and, at least in general, are not among the Killing vectors of the original undeformed background.

## 6.2 A trivial example

In the semi-simple setting, we are not aware of a Yang-Baxter model admitting enhanced symmetries, and suspect that they might not exist. In our current flat space setting however, the plane wave of the previous section provides an explicit example with enhanced symmetry, as we will come back to shortly. Before doing so, let us briefly discuss a trivial case, where the appearance of enhanced symmetries is obvious. Namely, consider the simple deformation of $\mathbb{R}^3$ generated by $r = p_1 \wedge p_2$. The corresponding background is given by

$$\begin{aligned} ds^2 &= \frac{1}{1 + \eta^2}(dx_1^2 + dx_2^2) + dx_3^2 \\ B &= -\frac{\eta}{1 + \eta^2}dx^1 \wedge dx^2 \end{aligned} \tag{6.10}$$

This deformation is trivial when considered for a closed string sigma model, since the $B$ field is constant and the metric is flat. This means this background admits full three dimensional Euclidean symmetry. At the same time, of the Killing vectors of the *undeformed* background, only $m_{12} = x_1\partial_2 - x_2\partial_1$ and $\partial_{1,2,3}$ remain, in line with the symmetries of the $r$ matrix. Geometrically

$$\mathcal{L}_{m_{13}}r = \mathcal{L}_{m_{13}}\partial_1 \wedge \partial_2 = \partial_3 \wedge \partial_2 \neq 0, \tag{6.11}$$

but – using two copies of the metric $g$ to turn this into a two form – we do have

$$d\left(g(\mathcal{L}_{m_{13}}r)g\right) = 0, \tag{6.12}$$

and similarly for $m_{23}$. Obviously, we can deform $m_{13}$ and $m_{23}$ to

$$\begin{aligned} \tilde{m}_{13} &= \sqrt{1 + \eta^2}^{-1}x_1\partial_3 - \sqrt{1 + \eta^2}x_3\partial_1 \\ \tilde{m}_{23} &= \sqrt{1 + \eta^2}^{-1}x_2\partial_3 - \sqrt{1 + \eta^2}x_3\partial_2 \end{aligned} \tag{6.13}$$

which are Killing vectors of the deformed metric, and complete our symmetry algebra. Relevantly,

$$\mathcal{L}_{\tilde{m}_{13}}B = -\frac{\eta}{\sqrt{1 + \eta^2}}dx^2 \wedge dx^3 \neq 0 \tag{6.14}$$

and similarly for $\tilde{m}_{23}$. Of course, we could simply drop the $B$ field here as it is constant, but in other models this is not always the case. Eqn. (6.14) shows that even the deformed symmetry generators $\tilde{m}_{13}$ and $\tilde{m}_{23}$ are *not* symmetries of the $r$ matrix. Moreover, $\tilde{m}_{13}$ and $\tilde{m}_{23}$ are not Killing vectors of the undeformed background.

## 6.3   The Nappi-Witten model and its extension

The situation for the Nappi-Witten background is more involved. The background (3.6) admits the following seven Killing vectors

$$
\begin{aligned}
&\chi_1 = \partial_+, &&\chi_2 = \partial_-, &&\chi_3 = x_3\partial_2 - x_2\partial_3,\\
&\chi_4 = \cos(\eta x^+)\partial_2 - \eta x_2\sin(\eta x^+)\partial_-, &&\chi_5 = \cos(\eta x^+)\partial_3 - \eta x_3\sin(\eta x^+)\partial_-, &&(6.15)\\
&\chi_6 = \sin(\eta x^+)\partial_2 + \eta x_2\cos(\eta x^+)\partial_-, &&\chi_7 = \sin(\eta x^+)\partial_3 + \eta x_3\cos(\eta x^+)\partial_-.
\end{aligned}
$$

where the $\eta$-independent $\chi_1, \chi_2, \chi_3$ correspond to the manifest symmetries of the Yang-Baxter model ($r$ matrix). The other four can be viewed as deformations of $p_2, p_3, m_{-2}, m_{-3}$, which they reduce to in the undeformed limit. The other three generators of $\mathfrak{iso}(1,3)$ of the undeformed model are fundamentally broken. When including the $B$ field, the Killing vector $\chi = \sum_i c_i\chi_i$ solves (6.9) with

$$
C = \sin(\eta x^+)\left(c_5 dx^2 - c_4 dx^3\right) - \cos(\eta x^+)\left(c_7 dx^2 - c_6 dx^3\right). \tag{6.16}
$$

As previously noted, the seven symmetries can be understood as left and right symmetries of the WZW action. Concretely we should identify

$$
\begin{aligned}
&P_1^L = \chi_4 - \chi_7, &&P_2^L = \chi_5 + \chi_6, &&J^L = \frac{\chi_1 - \eta\chi_3}{2} - b\chi_2,\\
&P_1^R = \chi_4 + \chi_7, &&P_2^R = \chi_5 - \chi_6, &&J^R = \frac{\chi_1 + \eta\chi_3}{2} - b\chi_2, &&(6.17)\\
&T = -2\chi_2,
\end{aligned}
$$

where the $L$ and $R$ superscripts denote the left and right copies of the symmetry algebra, with shared, hence unlabeled, central element $T$. These combinations of Killing vectors indeed have the expected commutation relations $[P_i, P_j] = \eta\epsilon_{ij}T$ and $[J, P_i] = \eta\epsilon_{ij}P_j$, independently for the left and right copies.[13]

Moving on, the six dimensional background (5.9) has the following twelve Killing vectors

$$
\begin{aligned}
&\chi_1 = \partial_+, &&\chi_2 = \partial_-, &&\chi_3 = x_3\partial_2 - x_2\partial_3,\\
&\chi_4 = \cos(\eta_1 x^+)\partial_2 - \eta_1 x_2\sin(\eta_1 x^+)\partial_-, &&\chi_5 = \cos(\eta_1 x^+)\partial_3 - \eta_1 x_3\sin(\eta_1 x^+)\partial_-,\\
&\chi_6 = \sin(\eta_1 x^+)\partial_2 + \eta_1 x_2\cos(\eta_1 x^+)\partial_-, &&\chi_7 = \sin(\eta_1 x^+)\partial_3 + \eta_1 x_3\cos(\eta_1 x^+)\partial_-, &&(6.18)\\
&\chi_8 = x_5\partial_4 - x_4\partial_5,\\
&\chi_9 = \cos(\eta_2 x^+)\partial_4 - \eta_2 x_4\sin(\eta_2 x^+)\partial_-, &&\chi_{10} = \cos(\eta_2 x^+)\partial_5 - \eta_2 x_5\sin(\eta_2 x^+)\partial_-,\\
&\chi_{11} = \sin(\eta_2 x^+)\partial_4 + \eta_2 x_4\cos(\eta_2 x^+)\partial_-, &&\chi_{12} = \sin(\eta_2 x^+)\partial_5 + \eta_2 x_5\cos(\eta_2 x^+)\partial_-.
\end{aligned}
$$

---

[13]Here we include the deformation parameter in the algebra, as in our discussion of the six dimensional analogue of the Nappi-Witten model, as opposed to the original Nappi-Witten conventions we used when discussing the four dimensional case earlier.

From the Yang-Baxter perspective only $\chi_1, \chi_2, \chi_3$ and $\chi_8$ are manifest symmetries. From the WZW model we expect $2d - 1 = 11$ isometries since there is one central element. The corresponding eleven generators correspond to the Killing vectors

$$P_1^{(1)L} = \chi_4 - \chi_7, \quad P_2^{(1)L} = \chi_5 + \chi_6, \quad P_1^{(2)L} = \chi_9 - \chi_{12}, \quad P_2^{(2)L} = \chi_{10} + \chi_{11},$$

$$P_1^{(1)R} = \chi_4 + \chi_7, \quad P_2^{(1)R} = \chi_5 - \chi_6, \quad P_1^{(2)R} = \chi_9 + \chi_{12}, \quad P_2^{(2)R} = \chi_{10} - \chi_{11},$$

$$J^L = \frac{\chi_1 - \eta_1 \chi_3 - \eta_2 \chi_8}{2} - b\chi_2, \quad J^R = \frac{\chi_1 + \eta_1 \chi_3 + \eta_2 \chi_8}{2} - b\chi_2, \quad T = -2\chi_2. \tag{6.19}$$

This leaves us with one remaining independent Killing vector – the antisymmetric combination $\eta_2\chi_3 - \eta_1\chi_8$ – not corresponding to a left or right $G$ symmetry generator of the WZW model. Moreover, in the special case of equal deformation parameters $\eta_1 = \eta_2$ the background admits a further two Killing vectors

$$\chi_{13} = x_4\partial_2 + x_5\partial_3 - x_2\partial_4 - x_3\partial_5, \qquad \chi_{14} = x_5\partial_2 - x_4\partial_3 + x_3\partial_4 - x_2\partial_5. \tag{6.20}$$

These three Killing vectors each correspond to an external automorphisms of the algebra defining our six dimensional WZW model. Namely, the six dimensional Nappi-Witten algebra admits two automorphisms corresponding to the independent rotations of the vectors $P^{(1)}$ and $P^{(2)}$, generated by

$$P_i^{(a)} \to \epsilon_i{}^j P_j^{(a)}, \qquad a = 1, 2. \tag{6.21}$$

These two automorphisms can be combined into the inner automorphism generated by $J$, and an independent external automorphism. Of course, any automorphism of $\mathfrak{g}$ acts simultaneously and identically on the left and right copies of $\mathfrak{g}$ in the symmetry algebra of the WZW model. The above two rotations of $P^{(1)}$ and $P^{(2)}$ are now precisely generated by $\chi_3$ and $\chi_8$, which enter in $J^L$ and $J^R$, and leave the independent combination $\eta_2\chi_3 - \eta_1\chi_8$ which generates the external automorphism. Next, for equal deformation parameters, the six dimensional algebra admits two further automorphisms, rotating between the two $P$ vectors. First, we have the external automorphism rotating the vectors $P_1$ and $P_2$, generated by

$$P_a^{(i)} \to \epsilon^i{}_j P_a^{(j)}, \qquad a = 1, 2. \tag{6.22}$$

which corresponds to the action of $\chi_{13}$ at the Killing vector level, again acting identically on both the left and right copies. Finally, we have the external automorphism rotating the vectors $V_1 = (P_1^{(1)}, P_2^{(2)})$ and $V_2 = (P_1^{(2)}, P_2^{(1)})$, generated by

$$V_a^i \to \epsilon^i{}_j V_a^j, \qquad a = 1, 2. \tag{6.23}$$

We see that upon taking external automorphisms into account, the WZW perspective manifests the full set of symmetries of the background also for this six dimensional model, as opposed to the Yang-Baxter perspective.[14] We believe the same applies to the higher dimensional versions of this model, but have not explicitly verified this.

---

[14] Of course for equal deformation parameters the existence of the two extra Killing vectors $\chi_{13}$ and $\chi_{14}$ is also manifest in the Yang-Baxter formulation, where they would rotate the two rotation generators appearing in the $r$ matrix into each other, which is a symmetry for equal deformation parameters.

## 6.4 An inhomogeneous deformation of the Nappi-Witten model

At this point we would briefly like to come back to the Yang-Baxter deformations of the Nappi-Witten model of [46] and [47], mentioned in footnote 9, as they provide another example of enhanced symmetry. Firstly however, let us come back to the apparent contradiction between the results of [46] and [47]. While the authors of [46] claim there is only one independent left Yang-Baxter deformation of the Nappi-Witten model, and that this only affects the relative coefficient of the $B$ field, the author of [47] claims to have found an inhomogeneous left Yang-Baxter deformation that interpolates from Nappi-Witten to flat space, given in eqn. (4.39) of section 4.5 of [47]. These results are in fact not contradictory, in the following sense. The deformed Nappi-Witten background of [47] is actually undeformed – it is diffeomorphic to the Nappi-Witten background. However, the Nappi-Witten model includes flat space in a particular limit, and in that sense there is space for (trivial) Yang-Baxter deformations which nevertheless interpolate between inequivalent models (Nappi-Witten and flat space). Concretely, in our conventions of eqs. (3.6),[15] it is clear that the Nappi-Witten model at $\eta = 0$ is actually flat space, while the models for any other value of $\eta$ are all equivalent, since any nonzero $\eta$ can be removed by rescaling $x^+$ and $x^-$ oppositely.

From the point of view of enhanced symmetries, the fact that there are a priori nontrivial Yang-Baxter deformations of the Nappi-Witten model [46] – i.e. ones associated to nonzero $r$ matrices, such as in particular the inhomogeneous $P_1 \wedge P_2$ deformation of eqs. (4.39) of [47] – which result in a trivial deformation of the actual model but breaks the original left $P_1$ and $P_2$ symmetry, means that also here we are dealing with enhanced symmetries. Like the $p_1 \wedge p_2$ deformation of flat space, these are trivial examples from a geometric point of view, but not from the point of view of the abstract Yang-Baxter model.

## 6.5 Abelian deformations of flat space

We have not determined the exact conditions under which a Yang-Baxter model admits enhanced symmetries, or what general form the corresponding Killing vectors take. As mentioned earlier, we are not aware of any semi-simple Yang-Baxter model admitting enhanced symmetries, essentially leaving us with our present setting of flat space, and the Nappi-Witten model just discussed.[16] To gain some insight into the type of $r$ matrices that allow for enhanced symmetries, and the form of the corresponding Killing vectors, we have checked all abelian rank two Yang-Baxter deformations of $\mathbb{R}^{1,3}$, summarizing our results in Tables 1 and 2. In addition to

---

[15]Our coordinates and ("opposite direction") deformation parameter are related to those of [47] as $\tilde{\eta} = \sqrt{1+2\eta}, u = x^+, v = x^- + \frac{\sqrt{1+2\eta}}{2}\left(x_2^2 + x_3^2\right) - \frac{b}{2}x^+, x = \sqrt{2+2\eta}\left(x_2\cos(x^+ + \eta x^+) - x_3\sin(x^+ + \eta x^+)\right), y = \sqrt{2+2\eta}\left(x_3\cos(x^+ + \eta x^+) + x_2\sin(x^+ + \eta x^+)\right)$, where we have denoted their deformation parameter as $\tilde{\eta}$.

[16]Left Yang-Baxter deformations of the Nappi-Witten model appear to give mainly geometrically trivial examples of enhanced symmetry, i.e. cases where the symmetry algebra is undeformed, although the $r$ matrix suggests otherwise. Viewed as trivial deformations of the background, or at most as a deformation of the coefficient of the $B$ field, they manifestly do not affect the symmetry algebra. Cases that can be viewed as "deforming" to flat space do give rise to nontrivial enhanced symmetries of course, as we go from a seven to a ten dimensional symmetry algebra. The latter case is just the reverse of our main discussion above, although from the Nappi-Witten perspective our particular deformation is a mixed left-right deformation rather than a purely left deformation of course. Other left-right deformations of the Nappi-Witten model may or may not give further interesting examples, but these would first need to be worked out along the lines of [53].

this we checked that the Yang-Baxter models for $r = m_{12} \wedge m_{34}$ and $r = m_{-2} \wedge m_{34}$ in five dimensions, have no enhanced symmetries.

| $r$ matrix | Manifest symmetries | Enhanced symmetries | Broken symmetries |
|---|---|---|---|
| $p_- \wedge p_+$ | $p_+, p_-, p_2, p_3, m_{23}, m_{+-}$ | $m_{+2}, m_{+3}, m_{-2}, m_{-3}$ | |
| $p_- \wedge p_2$ | $p_+, p_-, p_2, p_3, m_{-2}, m_{-3}$ | $m_{+2}, m_{+3}, m_{+-}, m_{23}$ | |
| $p_2 \wedge p_3$ | $p_+, p_-, p_2, p_3, m_{23}, m_{+-}$ | $m_{+2}, m_{+3}, m_{-2}, m_{-3}$ | |
| $p_- \wedge m_{-2}$ | $p_-, p_2, p_3, m_{-2}, m_{-3}$ | $p_+, m_{+2}, m_{+3}, m_{+-}, m_{23}$ | |
| $p_- \wedge m_{23}$ | $p_+, p_-, m_{23}$ | $p_2, p_3, m_{-2}, m_{-3}$ | $m_{+-}, m_{+2}, m_{+3}$ |
| $p_2 \wedge m_{+-}$ | $p_2, p_3, m_{+-}$ | | $m_{-3}, m_{+3}, m_{23}, p_+, p_-, m_{-2}, m_{-3}$ |
| $p_2 \wedge m_{-3}$ | $p_-, p_2, m_{-3}$ | $p_3, m_{-2}$ | $m_{+-}, m_{+2}, m_{+3}, p_+, m_{23}$ |
| $p_1 \wedge m_{23}$ | $p_+, p_-, m_{23}$ | | $m_{+-}, m_{+2}, m_{+3}, p_2, p_3, m_{-2}, m_{-3}$ |
| $m_{+-} \wedge m_{23}$ | $m_{+-}, m_{23}$ | | $p_+, p_-, p_2, p_3, m_{+2}, m_{+3}, m_{-2}, m_{-3}$ |
| $m_{-2} \wedge m_{-3}$ | $p_-, m_{-2}, m_{-3}, m_{23}$ | $p_2, p_3$ | $p_+, m_{+-}, m_{+2}, m_{+3}$ |

Table 1:    Overview of enhanced symmetries for abelian rank two deformations of $\mathbb{R}^{1,3}$ flat space. The enhanced symmetries are labeled by the undeformed generators admitting a suitable deformation to become enhanced symmetries. The broken symmetries column lists the generators which are fundamentally broken. The $r$ matrices are grouped by their length dimension in the Killing vector representation. The first four cases are maximally symmetric, i.e. correspond to undeformed flat space.

| $r$ matrix | Generator label | Deformed Killing vector |
|---|---|---|
| $p_- \wedge m_{23}$ | $p_2$ | $\cos(\eta x^+)\partial_2 - \eta x_2 \sin(\eta x^+)\partial_-$ |
| | $p_3$ | $\cos(\eta x^+)\partial_3 - \eta x_3 \sin(\eta x^+)\partial_-$ |
| | $m_{-2}$ | $\eta^{-1}\sin(\eta x^+)\partial_2 + x_2 \cos(\eta x^+)\partial_-$ |
| | $m_{-3}$ | $\eta^{-1}\sin(\eta x^+)\partial_3 + x_3 \cos(\eta x^+)\partial_-$ |
| $p_2 \wedge m_{-3}$ | $p_3$ | $\partial_3 - \eta^2 x^+ \left(x^+\partial_3 + x_3\partial_-\right)$ |
| | $m_{-2}$ | $x^2\partial_- + x^+\partial_2 + \frac{\eta^2}{3}(x^+)^3\partial_2$ |
| $m_{-2} \wedge m_{-3}$ | $p_2$ | $\partial_2 - \frac{\eta^2}{3}(x^+)^3\left(x^+\partial_2 + x_2\partial_-\right)$ |
| | $p_3$ | $\partial_3 - \frac{\eta^2}{3}(x^+)^3\left(x^+\partial_3 + x_3\partial_-\right)$ |

Table 2:   The Killing vectors corresponding to the enhanced symmetries of the nontrivial cases of Table 1.

## 6.6   Weyl symmmetry

In the process of investigating Yang-Baxter plane wave backgrounds, and looking for enhanced symmetries, we realized that there is another sense in which, at least in flat space, Yang-Baxter models can have enhanced symmetry. Namely, a Yang-Baxter deformed string sigma model is guaranteed to be one-loop Weyl invariant provided the $r$ matrix is unimodular [18] $- r^{ij}[t_i, t_j] = 0$ for $r = r^{ij}t_i \wedge t_j$ – and the only exceptions to unimodularity being a necessary condition for Weyl invariance were believed to be cases where the undeformed background $g + B$ is degenerate [54].

However, the Yang-Baxter deformation of flat space associated to

$$r = p_- \wedge m_{+-} \tag{6.24}$$

provides a counterexample to this. First of all, this jordanian type $r$ matrix is manifestly nonunimodular. Next, when we use it to deform flat space without a $B$ field (a nondegenerate starting point), the resulting background is nothing but flat space with zero $H$ flux again, which is certainly a Weyl invariant model. This example actually violates a subtle assumption underlying the analysis of [54], that the isometries involved in the $r$ matrix act without isotropy, which apparently allows for a non-unimodular but Weyl-invariant model, despite the non-degeneracy of $g + B$.[17]

Related to this, we would expect a non-unimodular $r$ matrix to give a background that solves the generalized supergravity equations [15, 16], which then generally would not solve the regular supergravity equations. In our case the Killing vector $K$ appearing in the generalized supergravity equations is presumably given by $K = \partial_-$, the Killing vector associated to the non-unimodularity of the $r$ matrix: $r|_{\wedge \to [,]} = p_-$.[18] Since in particular $K$ is null, this example satisfies the conditions for a trivial solution of generalized supergravity discussed around eqs. (4.5) in [17], which means we are effectively dealing with a solution of the regular supergravity equations, and hence a Weyl invariant model.[19]

We are not aware of other examples of non-unimodular but (manifestly) Weyl-invariant models in the present context, beyond trivial $p \wedge p$ extensions of the $r$ matrix (6.24).[20] We have checked that in four dimensions there are no other nonunimodular Yang-Baxter deformations which result in undeformed flat space.

# 7 Conclusions and outlook

We investigated plane wave backgrounds arising from Yang-Baxter deformations of the flat space string. For the simplest case with $r = p_- \wedge m_{23}$ this gives the so-called Nappi-Witten model, whose spectrum we determined by canonical quantization in light-cone gauge, and matched with an integrability-based approached based on a Drinfel'd twisted exact S matrix. In higher dimensions, it is possible to obtain analogues of the Nappi-Witten background as Yang-Baxter sigma models, for which the derivation of the spectrum by both methods follows similarly. Beyond explicitly verifying the quantum Drinfel'd twisted structure of this class of homogeneous deformations, the link of our Yang-Baxter models with Nappi-Witten type models shows that Yang-Baxter models can have more symmetries than those suggested by the deforming $r$ matrix. We illustrated this notion of enhanced symmetry for a number of abelian deformations of flat space. Finally, our investigations into plane waves and enhanced symmetries led us to realize that, at least for the non-semi-simple flat space string, there is at least one non-unimodular Yang-Baxter deformation which preserves Weyl invariance.

---

[17]We thank Linus Wulff for discussions on this point.

[18]The assumption that the isometries in the $r$ matrix act without isotropy is also made in [55], where the relation between $K$ and the $r$ matrix is given. We assume that this natural relation continues to apply here.

[19]We thank Riccardo Borsato for discussions on this point.

[20]The inhomogeneous $P_1 \wedge P_2$ Yang-Baxter deformation of the Nappi-Witten model is not unimodular, but in this case Weyl invariance is explained by the degeneracy of $g + B$ [47] in line with the analysis of [54].

There are a number of open questions directly associated to our results. Firstly, it would be interesting to study the deformed symmetry algebra of the Nappi-Witten model from the Yang-Baxter perspective, expected to take the shape of a Drinfel'd twisted Yangian, and determine how much of this can be explicitly seen at the quantum level. It would also be interesting to contrast this description with the original WZW CFT perspective on this model. Next, coming to enhanced symmetries, it would be great to determine exactly which type of Yang-Baxter models admits enhanced symmetries, in particular whether this could also arise in semi-simple models such as the $\text{AdS}_5 \times \text{S}^5$ string, and independently, to see if any enhanced symmetries present, admit an algebraic description from the Yang-Baxter perspective. Moreover, it would be interesting to see if the other examples of Yang-Baxter models with enhanced symmetries that we discussed, admit an alternative formulation that manifests these symmetries, similarly to the perspective provided by the WZW formulation in the Nappi-Witten case.[21] Finally, it would be great to strengthen the conditions for Weyl invariance of Yang-Baxter sigma models to an exact necessary requirement.

## Acknowledgments

We would like to thank Riccardo Borsato, Ben Hoare, Alessandro Sfondrini, and Linus Wulff for helpful discussions and Riccardo Borsato for comments on the draft. The work of the authors is supported by the German Research Foundation (DFG) via the Emmy Noether program "Exact Results in Extended Holography". ST is supported by LT.

---

[21] The other nontrivial abelian examples in Table 1 cannot be directly written as a WZW model for example, as a 4D WZW model has a symmetry algebra of dimension $8 - n$ where $n$ is the number of central elements, while from the Killing vectors it is easy to check that the examples do not have sufficiently many central elements.

# A    Quadratic worldsheet Hamiltonian

For the undeformed model one can fix the worldsheet reparameterization gauge freedom and avoid a square root worldsheet Hamiltonian by taking light-cone coordinates with metric

$$g_{\mu\nu} = \left( \begin{array}{cc|c} g_{++} & g_{+-} & 0 \\ g_{+-} & 0 & \\ \hline & 0 & g_{ij} \end{array} \right), \tag{A.1}$$

and fixing $x^+ = \tau, p_- = 1$, provided that the background is independent of $x^-$. We take the indices to run over $\mu \in \{+, -, i\}$. This will still be possible for the deformed model assuming that $r^{\mu+} = 0$, in this case we find

$$\mathcal{H}_{\mathrm{ws}} = -p_+ = \frac{1}{2g^{+-}} \left[ g^{ij} p_i p_j + g_{ij} \left( x'^i + \eta r^{i-} + \eta r^{ii'} p_{i'} \right) \left( x'^j + \eta r^{j-} + \eta r^{jj'} p_{j'} \right) + g^{--} \right]. \tag{A.2}$$

For flat space, one can obtain a worldsheet Hamiltonian that is at most quadratic in the dynamical fields, if $g^{--}$ is also at most quadratic and the remaining components $g^{+-}, g_{ij}$ are independent of dynamical variables. Here $x^+$ is not considered a dynamical variable and can appear arbitrarily, but would introduce world-sheet time dependence. Considering deformations of flat space in such a coordinate system and requiring it to remain quadratic after deformation, we need that $r^{i-}$ is at most linear and $r^{ij}$ is constant. In summary, to find a quadratic worldsheet Hamiltonian for a flat space deformation one should take Euclidean coordinates such that $g_{ij}$ are constant and the conditions on $r^{\mu\nu}$ are

$$r^{\mu+} = 0, \qquad \frac{\partial r^{\mu\nu}}{\partial x^-} = 0, \qquad \frac{\partial r^{ij}}{\partial x^k} = 0, \qquad \frac{\partial^2 r^{k-}}{\partial x^i \partial x^j} = 0. \tag{A.3}$$

## A.1    Plane wave $r$ matrix conditions

We are looking for a plane wave in light cone coordinates $\left( x^+, x^-, x^i \right)$. It is convenient to work with the inverse metric where non-zero components are

$$G^{--} = f(x), \qquad G^{+-} = -1, \qquad G^{ij} = \delta^{ij}. \tag{A.4}$$

For a Yang-Baxter deformation the inverse metric is given by

$$G^{\mu\nu} = g^{\mu\nu} - \eta^2 r^{\mu\mu'} g_{\mu'\nu'} r^{\nu'\nu}. \tag{A.5}$$

We now find constraints on $r^{\mu\nu}$, first by considering $G^{++} = g^{++} = 0$

$$G^{++} = g^{++} - \eta^2 r^{+\mu'} g_{\mu'\nu'} r^{\nu'+}, \tag{A.6}$$

$$r^{+\mu'} g_{\mu'\nu'} r^{\nu'+} = -\sum_i \left( r^{+i} \right)^2 = 0, \tag{A.7}$$

from this we may conclude that

$$r^{+i} = 0, \quad \forall i. \tag{A.8}$$

Next we consider $G^{+-} = g^{+-} = -1$

$$r^{+\mu'} g_{\mu'\nu'} r^{\nu'-} = r^{+-} g_{-+} r^{+-} + r^{+i} g_{ij} r^{j-} = 0, \tag{A.9}$$

using that $r^{+i} = 0$ we now conclude that

$$r^{+-} = 0. \tag{A.10}$$

And finally we may consider $G^{ii} = g^{ii} = 1$ using the results from before

$$r^{i\mu'} g_{\mu'\nu'} r^{\nu'i} = -\sum_j \left(r^{ij}\right)^2 = 0, \tag{A.11}$$

and conclude

$$r^{ij} = 0, \quad \forall i, j, \tag{A.12}$$

this means that the only non-zero components should be $r^{-i} = -r^{i-}$.

# B Plane wave background quantization

Before we quantize and construct the Fock space, we rescale the oscillators

$$\sqrt{4\pi\omega_n R}\, a_n^{\pm s} = \alpha_n^{\pm s}, \tag{B.1}$$

$$\sqrt{4\pi\omega_n R}\, \overline{a}_n^{\pm s} = \overline{\alpha}_n^{\pm s}. \tag{B.2}$$

It is worth noting, that for negative $\omega$ the $\alpha$'s look "antihermitian", $\alpha^* = -\overline{\alpha}$. With these definitions the Poisson brackets, energy and level matching condition becomes

$$\{\overline{\alpha}_n^{\pm}, \alpha_{n'}^{\pm}\} = \pm i\delta_{nn'}, \tag{B.3}$$

$$E = \sum_{n=-\infty}^{\infty} \omega_n \left(\overline{\alpha}_n^+ \alpha_n^+ + \overline{\alpha}_n^- \alpha_n^-\right), \tag{B.4}$$

$$L = \sum_{n=-\infty}^{\infty} n \left(\overline{\alpha}_n^+ \alpha_n^+ - \overline{\alpha}_n^- \alpha_n^-\right). \tag{B.5}$$

We quantize by replacing the Poisson bracket with a commutator

$$[\overline{\alpha}_n^{\pm}, \alpha_{n'}^{\pm}] = \pm\delta_{nn'}. \tag{B.6}$$

Let $\tilde{n} = \lceil n_0 \rceil$ where $n_0$ is the solution to $w_{n_0} = 0 = \frac{n_0}{R} + \eta$. To construct the Fock space we define the vacuum state as

$$\overline{\alpha}_{n\geq\tilde{n}}^+ |0\rangle = \alpha_{n\geq\tilde{n}}^- |0\rangle = \overline{\alpha}_{n<\tilde{n}}^- |0\rangle = \alpha_{n<\tilde{n}}^+ |0\rangle = 0, \tag{B.7}$$

Basis elements of the Fock space can be construced as

$$\left(\prod_{n=\tilde{n}}^{\infty} \left(\overline{\alpha}_n^-\right)^{p_n} \left(\alpha_n^+\right)^{q_n}\right) \left(\prod_{n=-\infty}^{\tilde{n}-1} \left(\overline{\alpha}_n^+\right)^{r_n} \left(\alpha_n^-\right)^{s_n}\right) |0\rangle = |\{p_n; q_n; r_n; s_n\}\rangle, \tag{B.8}$$

where $p_n, q_n, r_n, s_n \in \mathbb{N}^0$. The subset of physical states obey

$$L |\phi\rangle = 0, \tag{B.9}$$

$$L = \sum_{n=-\infty}^{\tilde{n}-1} n \left(\overline{\alpha}_n^+ \alpha_n^+ - \alpha_n^- \overline{\alpha}_n^-\right) + \sum_{n=\tilde{n}}^{\infty} n \left(\alpha_n^+ \overline{\alpha}_n^+ - \overline{\alpha}_n^- \alpha_n^-\right) \tag{B.10}$$

The normal ordered energy operator is

$$E = \sum_{n=-\infty}^{\tilde{n}-1} \omega_n \left( \overline{\alpha}_n^+ \alpha_n^+ + \alpha_n^- \overline{\alpha}_n^- \right) + \sum_{n=\tilde{n}}^{\infty} \omega_n \left( \alpha_n^+ \overline{\alpha}_n^+ + \overline{\alpha}_n^- \alpha_n^- \right). \tag{B.11}$$

Lets find the spectrum, we start by computing

$$E \left| \{p_n; q_n; r_n; s_n\} \right\rangle = E_{\{p,q,r,s\}} \left| \{p_n; q_n; r_n; s_n\} \right\rangle, \tag{B.12}$$

$$E_{\{p,q,r,s\}} = \sum_{n=-\infty}^{\tilde{n}-1} -\omega_n \left( r_n + s_n \right) + \sum_{n=\tilde{n}}^{\infty} \omega_n \left( p_n + q_n \right), \tag{B.13}$$

$$L \left| \{p_n; q_n; r_n; s_n\} \right\rangle = L_{\{p,q,r,s\}} \left| \{p_n; q_n; r_n; s_n\} \right\rangle, \tag{B.14}$$

$$L_{\{p,q,r,s\}} = \sum_{n=-\infty}^{\tilde{n}-1} n \left( r_n - s_n \right) + \sum_{n=\tilde{n}}^{\infty} n \left( q_n - p_n \right). \tag{B.15}$$

To write this in a more readable form we define

$$N_n = \begin{cases} q_n & n \geq \tilde{n} \\ r_n & n < \tilde{n} \end{cases}, \qquad\qquad \tilde{N}_n = \begin{cases} p_n & n \geq \tilde{n} \\ s_n & n < \tilde{n} \end{cases}. \tag{B.16}$$

Conceptually, they count left movers and right moving modes respectively. In terms of these integers the energy and level matching condition simply becomes

$$E_{\{N,\bar{N}\}} = \sum_{n=-\infty}^{\infty} \left| \frac{n}{R} + \eta \right| \left( N_n + \tilde{N}_n \right), \tag{B.17}$$

$$L_{\{N,\bar{N}\}} = \sum_{n=-\infty}^{\infty} n \left( N_n - \tilde{N}_n \right). \tag{B.18}$$

## C  Small deformation spectrum

The spectrum can be simply understood for small deformation parameter, we will restrict to $-\frac{1}{R} \leq \eta \leq \frac{1}{R}$. In this regime we can rewrite $\left| \frac{n}{R} + \eta \right| = \left| \frac{n}{R} \right| + \text{sign}(n) |\eta| + \delta_n^0 |\eta|$

$$E = \sum_{n \neq 0} \left( \left| \frac{n}{R} \right| + \text{sign}(n) |\eta| \right) \left( N_n + \tilde{N}_n \right) + |\eta| \left( N_0 + \tilde{N}_0 \right), \tag{C.1}$$

the level matching condition allows arbitrary $N_0, \tilde{N}_0$, this means we can add any integer factor of $|\eta|$. The minimum value of the bracket inside the sum happens at $n = -1$, this should come with an even integer factor due to level matching condition. This means the possible energy states are

$$E = \left( \frac{1}{R} - |\eta| \right) 2k_1 + |\eta| k_2, \tag{C.2}$$

for integers $k_1, k_2 \geq 0$ or slightly rewritten

$$E = \frac{2k}{R} + |\eta| \ell, \tag{C.3}$$

with integers $k \geq 0$ and $\ell \geq -2k$.

# D    Matrix representation for the Nappi-Witten algebra

The Nappi-Witten algebra spanned by $P_1, P_2, J, T$ with commutation relations

$$[J, P_i] = \epsilon_i{}^j P_j, \qquad [P_i, P_j] = \epsilon_{ij} T, \tag{D.1}$$

can be represented by the following matrices

$$P_1 = \begin{pmatrix} 0 & 1 & 0 & 0 \\ 0 & 0 & 0 & 0 \\ 0 & 0 & 0 & 0 \\ 0 & 0 & -1 & 0 \end{pmatrix}, \quad P_2 = \begin{pmatrix} 0 & 0 & 0 & 1 \\ 0 & 0 & 1 & 0 \\ 0 & 0 & 0 & 0 \\ 0 & 0 & 0 & 0 \end{pmatrix}, \quad J = \begin{pmatrix} 0 & 0 & 0 & 0 \\ 0 & 0 & 0 & -1 \\ 0 & 0 & 0 & 0 \\ 0 & 1 & 0 & 0 \end{pmatrix}, \quad T = \begin{pmatrix} 0 & 0 & 2 & 0 \\ 0 & 0 & 0 & 0 \\ 0 & 0 & 0 & 0 \\ 0 & 0 & 0 & 0 \end{pmatrix}. \tag{D.2}$$

We also provide a representation for the extended algebra (5.6) with $n = 2$,

$$P_1^{(1)} = \begin{pmatrix} 0 & \eta_1 & 0 & 0 & 0 & 0 \\ 0 & 0 & 0 & 0 & 0 & 0 \\ 0 & 0 & 0 & 0 & 0 & 0 \\ 0 & 0 & 0 & 0 & 0 & 0 \\ 0 & 0 & 0 & 0 & 0 & 0 \\ 0 & 0 & 0 & -\eta_1 & 0 & 0 \end{pmatrix}, \qquad P_2^{(1)} = \begin{pmatrix} 0 & 0 & 0 & 0 & 0 & 1 \\ 0 & 0 & 0 & 1 & 0 & 0 \\ 0 & 0 & 0 & 0 & 0 & 0 \\ 0 & 0 & 0 & 0 & 0 & 0 \\ 0 & 0 & 0 & 0 & 0 & 0 \\ 0 & 0 & 0 & 0 & 0 & 0 \end{pmatrix}, \tag{D.3}$$

$$P_1^{(2)} = \begin{pmatrix} 0 & 0 & \eta_2 & 0 & 0 & 0 \\ 0 & 0 & 0 & 0 & 0 & 0 \\ 0 & 0 & 0 & 0 & 0 & 0 \\ 0 & 0 & 0 & 0 & 0 & 0 \\ 0 & 0 & 0 & -\eta_2 & 0 & 0 \\ 0 & 0 & 0 & 0 & 0 & 0 \end{pmatrix}, \qquad P_2^{(2)} = \begin{pmatrix} 0 & 0 & 0 & 0 & 1 & 0 \\ 0 & 0 & 0 & 0 & 0 & 0 \\ 0 & 0 & 0 & 1 & 0 & 0 \\ 0 & 0 & 0 & 0 & 0 & 0 \\ 0 & 0 & 0 & 0 & 0 & 0 \\ 0 & 0 & 0 & 0 & 0 & 0 \end{pmatrix}, \tag{D.4}$$

$$J = \begin{pmatrix} 0 & 0 & 0 & 0 & 0 & 0 \\ 0 & 0 & 0 & 0 & 0 & -1 \\ 0 & 0 & 0 & 0 & -1 & 0 \\ 0 & 0 & 0 & 0 & 0 & 0 \\ 0 & 0 & \eta_2^2 & 0 & 0 & 0 \\ 0 & \eta_1^2 & 0 & 0 & 0 & 0 \end{pmatrix}, \qquad T = \begin{pmatrix} 0 & 0 & 0 & 2 & 0 & 0 \\ 0 & 0 & 0 & 0 & 0 & 0 \\ 0 & 0 & 0 & 0 & 0 & 0 \\ 0 & 0 & 0 & 0 & 0 & 0 \\ 0 & 0 & 0 & 0 & 0 & 0 \\ 0 & 0 & 0 & 0 & 0 & 0 \end{pmatrix}. \tag{D.5}$$

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
