# Peer review of "On exactly solvable Yang-Baxter models and enhanced symmetries"

_SciPost Physics_

## Round 2 · Referee Report · Anonymous (Referee 1) · 2024-6-5

Strengths

  1. The Nappi-Witten background was obtained from the flat space by the Yang-Baxter deformation, and the associated classical r-matrix was determined.

  2. Classical integrability of the Nappi-Witten model is argued via the Drinfeld twisted S-matrix.

  3. The symmetry enhancement of the Yang-Baxter deformed model was observed, which seems natural from the viewpoint of the local coordinates but non-trivial from that of the algebraic structures of the Yang-Baxter deformation.

  4. Presented an example of r-matrix, which is not unimodular but results in the Weyl invariant backgrounds.

Weaknesses

  1. The mechanism of symmetry enhancement is not entirely elucidated beyond exhibiting the examples.

  2. The reason why the r-matrix (6.24) shows the Weyl symmetry is not clearly explained. It would help readers if the authors wrote the resulting Weyl invariant backgrounds more explicitly.

  3. I am afraid that 29 footnotes are too much.

Report

This article investigates deformations of string sigma models, preserving the integrability of the original models, so-called Yang-Baxter sigma models. In particular, the authors have found the classical r-matrix that yields the Nappi-Witten background in Sec. 3.1 and its higher-dimensional generalization. The integrability is also argued in Sec. 4 based on the Drinfeld twist of the S-matrix. Then, the authors observed the symmetry enhancement of the Yang-Baxter deformed models in Sec. 6. Though the theoretical explanation of the enhancement is not completely solved, the authors listed such examples in Subsec. 6.5. In Subsec. 6.6, the non-unimodular r-matrix, which gives the Weyl invariant background, is discussed.
Since the results are significant in the context of the integrability of string background and its integrable deformations, I recommend this article for publication in this journal.

Requested changes

  1. In (2.2), mod 2 in the superscript seems strange, and I think it is better to remove it. Instead, for instance, the author could write a grading as $(0), (1) \in \mathbb{Z}/2\mathbb{Z}$.

  2. In Lax connection (2.9), $l_1(z)$ and $l_2(z)$ are not explicitly defined.

  3. In (2.13), the indices $\mu'$ and $\nu'$ do not seem canonically contracted in the first term.

  4. If footnote 13, the authors state "This appears to be at odds with our results showing that there is a Yang-Baxter deformation taking the Nappi-Witten model to flat space." If I could understand the argument of this paper correctly, I think the following comment is more precise; "This appears to be at odds with our results showing that there is a Yang-Baxter deformation taking flat space to the Nappi-Witten model." Because the starting background is the flat space, and the Nappi-Witten is the result. If so, I don't think this work is not at odds with [46].

Recommendation

Publish (meets expectations and criteria for this Journal)

---

## Round 2 · Referee Report · Anonymous (Referee 2) · 2024-7-26

Strengths

1 - Reformulation of the Nappi-Witten model as an integrable deformation of "Yang-Baxter" type of flat space.

2 - Confirmation of the Drinfeld twist interpretation of the deformation, by matching the canonical calculation of the spectrum with integrability-based methods.

3 - Observation that Yang-Baxter deformed models may possess extra symmetries compared to the ones usually identified in the literature.

4 - Identification of Yang-Baxter deformations that are compatible with Weyl invariance and that do not meet the sufficient conditions for Weyl invariance previously identified in the literature.

5 - Excellent and detailed presentation.

Weaknesses

1 - The authors focus on a restricted class of gravitational pp-wave backgrounds that they try to recast as Yang-Baxter deformations.

2 - The reason that gives rise to the enhancement of symmetries is left as an open question.

Report

The paper has very interesting results concerning the reinterpretation of a known model (Nappi-Witten) as a so-called "Yang-Baxter deformation" of flat space. The original motivation of the authors was to apply Yang-Baxter deformations to flat space to obtain backgrounds of a simple type (a certain class of gravitational pp-waves). These backgrounds can be quantised with canonical methods, so that the integrability-based proposal of implementing the deformation as a Drinfeld twist can be tested in a simple setup. Unfortunately, the resulting list of backgrounds of this type is limited. The authors perform a thorough study of the Yang-Baxter reinterpretation, and they identify unexpected extra symmetries of the models. The explanation for these extra symmetries is a non-trivial problem that is left open.

Recommendation

Publish (meets expectations and criteria for this Journal)

---

## Editorial Decision

resubmitted